# N-Terminal Pro-Brain Natriuretic Peptide Plasma Levels Are Associated with Intermediate-Term Follow-Up Cancer in Coronary Patients

**DOI:** 10.3390/jcm10184042

**Published:** 2021-09-07

**Authors:** José Tuñón, Ana Pello, Álvaro Aceña, Sergio Ramos-Cillán, Juan Martínez-Milla, Óscar González-Lorenzo, Jesús Fuentes-Antras, Nieves Tarín, Carmen Cristóbal, Luis M. Blanco-Colio, José Luis Martín-Ventura, Ana Huelmos, Carlos Gutiérrez-Landaluce, Marta López-Castillo, Joaquín Alonso, Lorenzo López Bescós, Jesús Egido, Ignacio Mahíllo-Fernández, Óscar Lorenzo

**Affiliations:** 1Department of Cardiology, IIS-Fundación Jiménez Díaz, 28040 Madrid, Spain; ampello@fjd.es (A.P.); aacena@fjd.es (Á.A.); juan.mmilla@fjd.es (J.M.-M.); ogonzalez@quironsalud.es (Ó.G.-L.); marta.lcastillo@fjd.es (M.L.-C.); 2Department of Medicine, School of Medicine, Autónoma University, 28040 Madrid, Spain; sergio.ramosc@quironsalud.es (S.R.-C.); jlmartin@fjd.es (J.L.M.-V.); jegido@fjd.es (J.E.); olorenzo@fjd.es (Ó.L.); 3Laboratory of Vascular Pathology, IIS-Fundación Jiménez Díaz, 28040 Madrid, Spain; lblanco@fjd.es; 4CIBERCV, 28040 Madrid, Spain; 5Department of Oncology, Hospital Clínico Universitario San Carlos, 28040 Madrid, Spain; jfuentesantras@outlook.com; 6Department of Cardiology, Hospital Universitario de Móstoles, 28040 Madrid, Spain; nieves.tarin@salud.madrid.org; 7Department of Cardiology, Hospital de Fuenlabrada, 28040 Madrid, Spain; carmen.cristobal@salud.madrid.org (C.C.); cgutierrezl@salud.madrid.org (C.G.-L.); 8Department of Medicine, School of Medicine, Rey Juan Carlos University, Alcorcón, 28040 Madrid, Spain; joaquinjalonso@gmail.com (J.A.); llbescos@secardiologia.es (L.L.B.); 9Department of Cardiology, Hospital Universitario Fundación Alcorcón, 28040 Madrid, Spain; ahuelmos@yahoo.es; 10Department of Cardiology, Hospital de Getafe, 28040 Madrid, Spain; 11CIBERDEM, 28040 Madrid, Spain; 12Research Unit, IIS-Fundación Jiménez Díaz, 28040 Madrid, Spain; imahillo@fjd.es

**Keywords:** coronary artery disease, N-terminal pro-brain natriuretic peptide, cancer, tumor, biomarker

## Abstract

N-terminal pro-brain natriuretic peptide (NT-proBNP) plasma levels are increased in patients with cancer. In this paper, we test whether NT-proBNP may identify patients who are going to receive a future cancer diagnosis (CD) in the intermediate-term follow-up. We studied 962 patients with stable coronary artery disease and free of cancer and heart failure at baseline. This sample represents a re-analysis of a previous work expanding the sample size and the follow-up. NT-proBNP, galectin-3, monocyte chemoattractant protein-1, high-sensitivity C-reactive protein, high-sensitivity cardiac troponin I (hsTnI), and calcidiol (vitamin D) plasma levels were assessed. The primary outcome was new CD. After 5.40 (2.81–6.94) years of follow-up, 59 patients received a CD. NT-proBNP [HR 1.036 CI (1.015–1.056) per increase in 100 pg/mL; *p* = 0.001], previous atrial fibrillation (HR 3.140 CI (1.196–8.243); *p* = 0.020), and absence of previous heart failure (HR 0.067 CI (0.006–0.802); *p* = 0.033) were independent predictors of receiving a CD in the first three years of follow-up. None of the variables analyzed predicted a CD beyond this time. The number of patients developing heart failure during follow-up was 0 (0.0%) in patients receiving CD in the first three years of follow-up, 2 (6.9%) in those receiving a CD diagnosis beyond this time, and 40 (4.4%) in patients not developing cancer (*p* = 0.216). These numbers suggest that future heart failure was not a confounding factor. In patients with coronary artery disease, NT-proBNP was an independent predictor of CD in the first three years of follow-up but not later, suggesting that it could be detecting subclinical undiagnosed cancers.

## 1. Introduction

Patients with coronary artery disease (CAD) are at risk of developing malignancies, given that cancer shares some risk factors with this disorder, such as age, smoking, and even some dietary patterns [1,2,3]. Thus, finding biomarkers that predict both the risk of cancer and of cardiovascular events could be useful in CAD patients.

Natriuretic peptides are secreted by cancer cells [4,5] and N-terminal fragments of pro-brain natriuretic peptide (NT-proBNP) levels are increased in patients with cancer [6]. Previously we have reported that NT-proBNP plasma levels predict a future diagnosis of cancer in 704 patients with CAD from Biomarkers in Acute Coronary Syndrome & Biomarkers in Acute Myocardial Infarction (BACS & BAMI) studies [7]. We launched the hypothesis that NT-proBNP could be merely detecting subclinical cancers rather than being a real predictor.

In the present work, we present a re-analysis of patients from BACS & BAMI studies, expanding the sample size and follow-up, and studying 962 patients with CAD free of malignancies at baseline.

In this paper, we test the hypothesis that NT-proBNP plasma levels predict a new cancer diagnosis (CD) only in the intermediate-term follow-up but not in the long-run. Along with NT-proBNP, we also tested these biomarkers: monocyte chemoattractant protein-1 (MCP-1), involved in inflammation and atherothrombosis, among other processes [8,9]: galectin-3, related to malignancies, heart failure, thrombosis, and renal dysfunction [10,11]; high-sensitivity cardiac troponin I (hsTnI), which has been described to have prognostic value in stable CAD [12]; and vitamin D (calcidiol) plasma levels, as low levels of this molecule were related to cancer [13]. High-sensitivity C-reactive protein (hsCRP) was studied as a reference, given the large amount of information published on this biomarker on cardiovascular disease.

## 2. Materials and Methods

### 2.1. Patients

Nine-hundred and sixty-eight patients with stable CAD, who had suffered acute coronary syndrome between six and twelve months before were included in this study. These patients were part of the BACS & BAMI (Biomarkers in Acute Coronary Syndrome & Biomarkers in Acute Myocardial Infarction) studies, carried out in five hospitals in Madrid. Inclusion and exclusion criteria were defined previously [14].

The research protocol conforms to the ethical guidelines of the 1975 Declaration of Helsinky as reflected in a priori approval by the human research committees of the institutions participating in this study: Fundación Jiménez Díaz, Hospital Fundación Alcorcón, Hospital de Fuenlabrada, Hospital Universitario Puerta de Hierro Majadahonda, and Hospital Universitario de Móstoles. All patients signed informed consent documents. Patients and public were not involved in the design of this study.

### 2.2. Study Design

At baseline, clinical variables were recorded, and twelve-hour fasting venous blood samples were withdrawn and collected in EDTA. Blood samples were centrifuged at 2500× *g* for 10 min and plasma was stored at −80 °C. Patients were seen every year at their hospital. At the end of follow-up, the medical records were reviewed, and patient status was confirmed by telephone contact. Plasma extraction and baseline visits took place between January 2007 and December 2014. The last follow-up visits were carried out in June 2016.

The primary outcome was the development of a new cancer with histological confirmation, excluding non-melanocytic skin tumors.

### 2.3. Analytical Studies

Blood samples were collected in ethylene-diamine-tetra-acetic acid and centrifuged at 2500× *g* for 10 min and plasma was stored at −80 °C. Patients were seen every year at their hospital.

The investigators who performed the laboratory determinations were unaware of the clinical data. Plasma concentrations of MCP-1 and galectin-3 were determined in duplicate using commercially available enzyme-linked immunosorbent assay kits (DCP00, R&D Systems and BMS279/2 Bender MedSystems, respectively) following the manufacters’ instructions. HsCRP was assessed by latex-enhanced immunoturbidimetry (ADVIA 2400 Chemistry System, Siemens, Munich, Germany), NT-proBNP by immunoassay (VITROS, Orthoclinical Diagnostics, Raritan, NJ, USA), and hsTnI was assessed by direct quimioluminiscence (ADVIA Centaur, Siemens, Munich, Germany). Plasma calcidiol levels were quantified by chemiluminescent immunoassay (CLIA) on the LIAISON XL analyzer (LIAISON 25OH-Vitamin D total Assay DiaSorin, Saluggia, Italy). Lipids, glucose, and creatinin were determined by standard methods (ADVIA 2400 Chemistry System, Siemens, Germany).

### 2.4. Statistical Analysis

Quantitative data following a normal distribution are displayed as mean ± standard deviation and compared using the Student *t*-test and the ANOVA (analysis of variance) for two and three comparisons, respectively. Quantitative data not following a normal distribution are displayed as median (interquartile range) and compared using the Mann-Whitney and the Kruskal-Wallis tests for two and three comparisons, respectively. Qualitative variables are displayed as percentages and compared by χ^2^ or the Fisher exact test when appropriate. Univariate Cox regression analysis was used to study the predictive power of the variables studied. Multivariate Cox regression analysis was performed including all variables showing *p* < 0.20 at univariate analysis.

Analyses were performed with R v3.5.1 (R foundation for Statistical Computing, Viena, Austria) and SPSS 19.0 (SPSS Inc., New York, NY, USA). Statistical significance was set at *p* < 0.05 (two-tailed).

## 3. Results

### 3.1. Population of the Study

We included 968 patients with stable CAD. However, five patients were lost to follow-up, and one patient died, and the cause of death could not be confirmed. Thus, he was also considered as lost to follow-up, leaving 962 patients for the analysis, who were followed for an average period of 5.40 (2.81–6.94) years.

Of the 962 patients studied, 59 developed cancer during follow-up. We divided the sample into three subgroups: patients who did not develop cancer during follow-up (*n* = 903), those receiving a CD in the first 3 years of follow-up (*n* = 30), and patients receiving a CD beyond 3 years (*n* = 29). We used the three-year cutt-off as it divided the sample in a similar number of patients receiving a CD before and beyond this point. Overall, baseline variables were well balanced among the three subgroups of patients, and the only significant differences observed were: more incidence of previous heart failure in patients not receiving a CD during follow-up, higher hsCRP levels in those receiving a CD in the first three years of follow-up, and fewer patients receiving P2Y12 inhibitors in the group receiving a CD beyond three years of follow-up (Table 1). There were no significant differences in in the presence of left-systolic ventricular dysfunction, diabetes or hypertension, among others.

In the group of patients receiving a CD in the first three years, the median time to CD was 1.53 (0.50–2.35) years. In this group, there were 15 (50.0%) carcinomas, 11 (36.7%) adenocarcinomas, 3 (10.0%) cancers with other histology, and 1 (3.3%) case of unknown etiology. Among patients receiving a CD beyond this time the median time to a CD was 4.98 (3.12–9.28) years. In this group, there were 13 (44.8%) carcinomas, 11 (37.9%) adenocarcinomas, 5 (17.2%) cancers with other histology, and 0 (0.0%) cancers of unknown histology. The case of unknown histology corresponded to an 83-year-old man with images clearly compatible with lung cancer with metastases, who was dying, and the doctors decided not to perform histological analysis. Table 2 shows the locations of the cancers.

### 3.2. NT-proBNP Plasma Levels Predict a Future Diagnosis of Cancer

We performed a univariate Cox analysis for all the variables displayed in Table 1, and the result is shown in Table 3. Only age, previous atrial fibrillation, NT-proBNP, and hs-Troponin I plasma levels were associated with the development of cancer in the three first years of follow-up. No variable was associated with a cancer diagnosis beyond that time.

Multivariate Cox regression was performed including variables that showed a *p* value < 0.20 at the univariate analysis. We chose this cut-off instead, 0.05, given that only four variables had a *p* value < 0.05, and important information could have been lost if we only included these variables in the multivariate model. NT-proBNP was a strong, independent predictor of developing cancer in the first three years of follow-up, along with the existence of previous atrial fibrillation and the absence of previous heart failure (Table 4). There were no independent predictors of developing cancers beyond three years of follow-up.

Forty patients (4.4%) developed heart failure during follow-up in the group not receiving a CD, 0 (0.0%) in patients receiving a CD during the first three years of follow-up, and 2 (6.9%) in patients receiving a CD beyond three years (*p* = 0.216).

### 3.3. NT-proBNP Plasma Levels in Different Types of Cancers Diagnosed before Three Years of Follow-Up

In the subgroup of patients developing cancer in the first 3 years, NT-proBNP levels were 519.8 ± 706.5 pg/mL (N = 15) in those with carcinoma and 480.4 ± 143.0 pg/mL (*n* = 11) in patients developing adenocarcinoma (*p* = 0.877). Lung and hematologic cancers (*n* = 6) did not have NT-proBNP levels significantly different to other tumors (*n* = 24) (428.5 (152.3, 1212.0) vs. 257.5 (88.0, 981.3) pg/mL respectively; *p* = 0.402).

## 4. Discussion

Patients with CAD have high probabilities of developing tumors, given that the incidence of cancer increases with age [1], tobacco consumption, and some dietary patterns that also promote CAD [2,3]. Thus, predicting the risk of cancer in this population could mean an improvement in the assessment of their global prognosis.

NT-proBNP is mainly used in diagnosing heart failure [15], although it may also predict the development of heart failure and death in patients with cardiovascular disease [14,16]. In addition, it has been associated with total death in elderly subjects [17,18,19,20].

In 2015, we described for the first time that NT-proBNP is an independent predictor of the appearance of malignancies in patients with CAD [7]. However, we hypothesized that NT-proBNP levels could probably be a marker of subclinical tumors rather than a real predictor of the development of new cancers. To test this hypothesis, we have expanded the sample size and extended the follow-up period of the same series of patients, reporting a total of 59 patients developing cancer instead of the 24 cases reported previously. Then, we divided the patients receiving a new cancer diagnosis during the follow-up into those receiving this diagnosis before and after three years. NT-proBNP was an independent predictor of developing a cancer in the first three years of follow-up, but not later.

The likelihood that concomitant heart failure may have influenced our results is very slim. First, the presence of previous heart failure was very low in the whole sample and significantly lower in patients developing a cancer in the first three years as compared with patients with no cancer at follow-up. Second, a low percentage of our patients developed heart failure during follow-up, without significant differences between the cancer and non-cancer groups. Interestingly, no patient in the group receiving a cancer diagnosis in the first three years developed heart failure during follow-up. Also, variables that may influence NT-proBNP levels, such as age, sex, hypertension, atrial fibrillation, glomerular filtration rate and body-mass index [16,21,22] were included in the multivariate analysis, limiting the possibility that they could have influenced the results.

Natriuretic peptides are used in patients with cancer mainly as predictors of cardiac toxicity secondary to chemotherapy [23]. However, a potential relationship between plasma levels of these peptides and cancer itself has been suggested, and patients with cancer may have elevated BNP levels in the absence of heart failure [6,24]. In patients >40 years of age with previous non-cardiac surgery, NT-proBNP levels ≥ 125 pg/mL were independently associated with lung cancer after excluding cases with heart failure, CAD, and other conditions known to affect this biomarker [25]. Although natriuretic peptides were shown to be secreted by small-cell lung cancer [5], only 15% of the cases reported in this previous study, and around 17% in the present paper, had this type of cancer, suggesting that other tumors could also produce NT-proBNP. In addition, NT-proBNP was an independent predictor of survival in patients with non-Hodgkin lymphoma [26]. Interestingly, NT-proBNP levels were associated with the involvement of two or more extranodal sites, suggesting a potential relationship with the stage of this malignancy. Similarly, increased NT-proBNP plasma levels predict the progression and worse outcome of metastatic renal carcinoma [27,28]. In this regard, elevated NT-proBNP levels were reported to be associated with total mortality in patients with cancer and no previous cardiotoxic anticancer therapy who were stable from a cardiovascular point of view [29]. Additionally, NT-proBNP has been found in some series to be of value guiding therapy in heart failure. However, this effect was lost in patients with cancer and other comorbidities, suggesting that NT-proBNP levels may be influenced by this condition [30]. Finally, recent data suggest that cancer biomarkers correlate positively with NT-proBNP plasma levels in patients with heart failure, and even some of them were noninferior to NT-proBNP in predicting all-cause mortality [31]. These findings suggest a bidirectional relationship between biomarkers of cancer and heart failure.

Cancer cells may produce natriuretic peptides. Small-cell lung cancer may secrete both pro-atrial natriuretic peptide and BNP [4,5]. Also, BNP is expressed in normal adrenal glands and in adrenal tumors [32]. NT-proBNP synthesis may be stimulated by several proinflammatory cytokines [33] that, for instance, are expressed in Hodgkin lymphoma [34,35]. Moreover, these cytokines may predict clinical outcome in diffuse large B-cell lymphomas [36,37,38] and are increased in malignancies at advanced stages [26].

The specific cause of the elevation of natriuretic peptide plasma levels seen in cancer has not yet been elucidated. It has been demonstrated that these peptides decrease the number of several cancer-cell types in vitro through a reduction of DNA synthesis [39] and inhibition of c-Fos and c-Jun protooncogenes [40]. They also diminish the expression of vascular endothelial growth factor and that of its receptor VEGFR2, thus suggesting that these peptides have the potential to control vasculogenesis [41]. One work has shown opposite effects of natriuretic peptides on carcinogenesis depending on their concentrations [42]. Overall, natriuretic peptides seem to decrease the proliferation of cancer cells. Accordingly, they inhibit lung metastases and skin carcinogenesis in animal models [43,44].

Given that most data suggest an anti-cancer effect of natriuretic peptides, it is plausible that their production by cancer cells represents a negative feed-back mechanism trying to control tumor growth. In our study, NT-proBNP was only useful in predicting tumors that were diagnosed in the first three years of follow-up, losing its predictive power beyond this time. This supports the hypothesis that NT-proBNP detects tumors that are present subclinically at the moment of blood extraction and which are not evident using the tools currently available in clinical practice. In that case, NT-proBNP could be useful for the early detection of malignancies, as not all cancers described in this paper have a specific biomarker to aid in diagnosis.

This work has certain limitations: (1) The inverse association between previous heart failure and future cancer must be interpreted with caution, due to the low percentage of patients with heart failure in addition to the limited number of them who developed cancer during follow-up; (2) Given the relatively low number of patients who developed cancer, these findings should be confirmed in other studies, probably including populations without heart failure at baseline; (3) Additionally, given the limited number of patients who developed cancer we could not establish whether the predictive effect of NT-proBNP is restricted to some types of cancer or can be applied to any malignancy; (4) Having repeated NT-proBNP levels during follow-up could have aded consistency to the data, but taking more plasma samples from patients was not included in the protocol of the study.

## 5. Conclusions

NT-proBNP might be an independent predictor of malignancies in the intermediate-term follow up in patients with stable CAD. Further prospective studies with larger populations are needed to validate these results.

## Figures and Tables

**Table 1 jcm-10-04042-t001:** Baseline characteristics of the patients.

Characteristic	Patients without Cancer (*n* = 903)	Patients with Cancer in 3 Years (*n* = 30)	Patients with Cancer beyond 3 Years (*n* = 29)	*p* Value
Age (yr)	61.1 ± 12.0	66.0 ± 11.6	62.2 ± 12.3	0.083
Male sex (%)	23.5	30.0	27.6	0.632
Body-mass index (Kg/m^2^)	28.5 ± 4.35	27.8 ± 4.71	28.8 ± 3.78	0.621
Diabetes (%)	24.0	16.7	31.0	0.434
Present smoker (%)	14.0	13.3	10.3	0.955
Present or past smoker (%)	75.6	86.7	75.9	0.380
Hypertension (%)	63.5	80.0	72.4	0.115
Previous heart failure (%)	12.3	3.3	0.0	0.036
Peripheral artery disease (%)	3.8	3.3	3.4	1.000
Cerebrovascular events (%)	2.9	0.0	3.4	0.660
Ejection fraction < 40% (%)	6.9	6.9	10.7	0.585
Present or past atrial fibrillation (%)	6.0	16.7	6.9	0.064
Medical therapy				
Acetylsalicylic acid (%)	92.6	96.7	89.7	0.556
AntiP2Y12 (%)	75.7	76.7	51.7	0.013
Acenocumarol (%)	5.2	3.3	3.4	1.000
Statins (%)	94.5	93.3	93.1	0.567
Oral antidiabetic drugs (%)	16.9	10.0	24.1	0.351
Insulin (%)	7.0	0.0	0.0	0.150
ACEI (%)	62.5	63.3	44.8	0.155
Angiotensin receptor blockers (%)	15.0	13.3	24.1	0.370
Aldosterone receptor blockers (%)	7.0	6.7	3.4	0.919
Betablockers (%)	78.8	73.3	65.5	0.184
Diuretics (%)	18.8	23.3	27.6	0.422
Amiodarone	0.9	0.0	0.0	1.000
Digoxine	0.3	0.0	0.0	1.000
ANALYTICAL DATA				
LDL cholesterol (mg/dL)	80.5 ± 25.0	78.1 ± 25.2	85.3 ± 22.5	0.518
HDL cholesterol (mg/dL)	42.0 ± 10.8	43.4 ± 10.7	44.5 ± 13.6	0.397
Non-HDL cholesterol	104 ± 30.5	98.4 ± 27.1	111 ± 28.2	0.291
Triglycerides (mg/dL)	101 (66.0)	92.5 (66.2)	121 (56.0)	0.135
Glycemia (mg/dL)	101 (24.0)	98.0 (17.8)	99.0 (19.0)	0.792
eGFR (CKD-EPI) (ml/min/1.73 m^2^)	78.6 ± 19.4	74.4 ± 19.2	73.1 ± 17.8	0.171
Hs C-reactive protein (mg/L)	1.08 (2.72)	2.68 (4.63)	1.49 (2.81)	0.004
NT-ProBNP (pg/mL)	176 (297)	272 (815)	155 (217)	0.324
MCP-1 (pg/mL)	135 (74)	151 (58)	133 (43)	0.725
Galectin-3 (ng/mL)	7.86 (3.89)	7.85 (4.13)	7.50 (2.89)	0.253
Hs troponin I (ng/mL)	0.003 (0.010)	0.002 (0.016)	0.003 (0.009)	0.837
Calcidiol (ng/mL)	20.4 ± 8.55	21.4 ± 9.8	17.6 ± 6.93	0.175

Values are presented as mean ± standard deviation and median (interquartile range). ACEI: angiotensin-converting inhibitors; CKD-EPI: chronic kidney disease epidemiology collaboration equation; eGFR: estimated glomerular filtration rate; HDL: high-density lipoprotein; Hs: high-sensitivity; LDL: low-density lipoprotein MCP-1: monocyte chemoattractant protein-1; NT-Pro-BNP: N-terminal pro-Brain natriuretic peptide.

**Table 2 jcm-10-04042-t002:** Location of the cancers diagnosed during follow-up.

Cancer Location	Cancer < 3 Years	Cancer > 3 Years
Prostate	5 (16.7%)	6 (20.7%)
Liposarcoma	1 (3.3%)	0 (0%)
Esophagus	1 (3.3%)	1 (3.4%)
Pancreas	2 (6.7%)	1 (3.4%)
Melanoma	1 (3.3%)	1 (3.4%)
Pharynx and mouth	1 (3.3%)	2 (6.9%)
Uterus	0 (0%)	1 (3.4%)
Liver and biliary system	0 (0%)	1 (3.4%)
Colon	3 (10%)	3 (10.3%)
Lung	5 (16.7%)	5 (17.2%)
Leukemia	0 (0%)	1 (3.4%)
Larynx	3 (10%)	0 (0%)
Urinary bladder/ureter	2 (6.7%)	1 (3.4%)
Breast	2 (6.7%)	2 (6.9%)
Lymphoma	1 (3.3%)	1 (3.4%)
Kidney	3 (10%)	3 (10.3%)

**Table 3 jcm-10-04042-t003:** Univariate analysis of cancer predictors before and after three years of follow-up.

	Cancer before 3 Years			Cancer after 3 Years		
	HR	(95% CI)	*p*	HR	(95% CI)	*p*
Age	1.034	(1.003, 1.065)	0.032	1.009	(0.978, 1.041)	0.583
Sex (woman)	0.753	(0.345, 1.644)	0.476	0.830	(0.366, 1.886)	0.657
Diabetes	0.640	(0.245, 1.672)	0.363	1.257	(0.553, 2.855)	0.585
Body-mass index	0.956	(0.872, 1.049)	0.342	0.999	(0.915, 1.089)	0.976
Past or present smoker	2.150	(0.750, 6.160)	0.154	1.072	(0.455, 2.523)	0.874
Present smoker	0.981	(0.342, 2.811)	0.971	0.602	(0.143, 2.536)	0.489
Hypertension	2.183	(0.892, 5.340)	0.087	1.362	(0.600, 3.092)	0.461
Previous heart failure	0.262	(0.036, 1.925)	0.188			
Peripheral artery disease	0.911	(0.124, 6.689)	0.927	0.983	(0.133, 7.260)	0.987
Cerebrovascular events				1.354	(0.184, 9.98)	0.766
Ejection fraction < 40%	0.971	(0.231, 4.085)	0.968	1.794	(0.540, 5.962)	0.340
Present or past atrial fibrillation	3.144	(1.203, 8.216)	0.019	1.439	(0.340, 6.082)	0.621
Acetylsalicylic acid	2.476	(0.337, 18.18)	0.373	0.843	(0.254, 2.793)	0.780
AntiP2Y12	1.156	(0.496, 2.695)	0.737	0.498	(0.237, 1.048)	0.066
Acenocumarol	0.667	(0.091, 4.895)	0.690	0.685	(0.093, 5.042)	0.710
Statins	0.899	(0.214, 3.776)	0.885	0.902	(0.214, 3.801)	0.888
Oral antidiabetic drugs	0.559	(0.170, 1.844)	0.340	1.299	(0.526, 3.208)	0.570
Insulin						
ACEI	1.133	(0.539, 2.381)	0.743	0.757	(0.360, 1.593)	0.463
Angiotensin-receptor blockers	0.847	(0.296, 2.428)	0.758	1.355	(0.549, 3.345)	0.509
Aldosterone receptor blockers	1.031	(0.245, 4.328)	0.967	0.694	(0.094, 5.109)	0.720
Betablockers	0.782	(0.348, 1.758)	0.552	0.687	(0.310, 1.518)	0.353
Diuretic	1.264	(0.542, 2.945)	0.588	1.319	(0.560, 3.104)	0.526
Amiodarone						
Digoxin LDL	0.994	(0.978, 1.009)	0.419	1.002	(0.988, 1.017)	0.776
HDL	1.006	(0.974, 1.039)	0.710	1.010	(0.978, 1.042)	0.554
Non-HDL	0.991	(0.977, 1.004)	0.172	1.001	(0.990, 1.013)	0.819
Triglycerides	0.994	(0.986, 1.001)	0.105	0.999	(0.994, 1.004)	0.734
Glucose	1.001	(0.990, 1.012)	0.879	1.006	(0.999, 1.013)	0.098
eGFR	0.991	(0.973, 1.008)	0.301	0.988	(0.969, 1.008)	0.233
Hs C-Reactive protein	1.007	(0.977, 1.038)	0.642	0.990	(0.937, 1.045)	0.709
NT-proBNP *	1.020	(1.004, 1.035)	0.012	0.987	(0.922, 1.057)	0.712
MCP-1	0.797	(0.434, 1.463)	0.464	0.750	(0.365, 1.541)	0.434
Galectin-3	0.983	(0.888, 1.087)	0.734	0.910	(0.796, 1.040)	0.165
Hs-Troponin I ^#^	1.052	(1.005, 1.102)	0.029	1.516	(0.704, 3.265)	0.287
Calcidiol	1.017	(0.977, 1.058)	0.418	0.969	(0.920, 1.019)	0.222

* Risk change per 100 units of increment in plasma levels. ^#^ Risk change per 0.1 units of increment in plasma levels. Variables without results correspond to conditions that were not present in the subgroups described. HR = hazard ratio; CI = confidence interval; other abbreviations as for Table 1.

**Table 4 jcm-10-04042-t004:** Multivariate analysis showing the predictors of a cancer diagnosis before 3 years of follow-up.

	Hazard Ratio (95% CI)	*p* Value
* **Variables included in the final model** *		
NT-proBNP *	1.036 (1.015, 1.056)	0.001
Atrial fibrillation	3.140 (1.196, 8.243)	0.020
Previous heart failure	0.067 (0.006, 0.802)	0.033
* **Variables not included in the final model** *		
Age	1.024 (0.992, 1.057)	0.140
Past or present smoker	2.121 (0.739, 6.090)	0.162
Hypertension	2.006 (0.810, 4.968)	0.132
Non-HDL cholesterol	0.992 (0.979, 1.005)	0.238
Triglycerides	0.994 (0.987, 1.002)	0.149
Hs-Troponin I **^#^**	1.031 (0.981, 1.084)	0.227

CI: Confidence Interval; NT-proBNP: N-terminal pro-brain natriuretic peptide. * Risk change per 100 units of increment in plasma levels. ^#^ Risk change per 0.1 units of increment in plasma levels.

## Data Availability

Data are available on request to the corresponding author.

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
