# Peer review of "N-Terminal Pro-Brain Natriuretic Peptide Plasma Levels Are Associated with Intermediate-Term Follow-Up Cancer in Coronary Patients"

_jcm, 2021, doi:10.3390/jcm10184042_

Round 1

Reviewer 1 Report

The authors extended their prior analysis to show in this paper that now 59/962 patients on the BACS and BAMI studies developed a new cancer diagnosis (vs 24/699 on initial publication). These patients at baseline had stable CAD, and no heart failure or known cancer diagnosis. Importantly, with longer follow-up and more events, the authors observed that NT-proBNP, previous atrial fibrillation, and absence of prior heart failure were independent predictors of a subsequent cancer diagnosis within 3 years. None of the evaluated factors predicted for cancer diagnosis after 3 years. This is an important study that adds identifies NT-proBNP as worthy of further study for validation and as a novel putative cancer biomarker in patients with CAD.

My specific comments are:

  1. The abstract methods should describe that this is a re-analysis of patients that were on a prospective trial
  2. Why was 3 years chosen?
  3. What was the median time/interval to developing cancer? (in the <3 yr and >3 yr cohorts)?
  4. While there wasn’t a statistically significant difference in the subgroup comparisons by histology for NT-proBNP levels (e.g. adeno vs carcinoma, lung/hem vs other), it would be really interesting to see the mean NT-proBNP levels for the various cancer sites/histologies in a table to assess trends. It might also be interesting to look at the squamous cell vs adeno subsets.
  5. Table 4 should include the HR/95% CIs/p-values for all the variables included in the model.  

Author Response

ANSWERS FOR REVIEWER  1

First of all we want to thank the reviewers and the editors for their comments that have undoubtedly contributed to improve the quality of the manuscript.

Below, we provide our answers to the reviewers.

REVIEWER 1

The authors extended their prior analysis to show in this paper that now 59/962 patients on the BACS and BAMI studies developed a new cancer diagnosis (vs 24/699 on initial publication). These patients at baseline had stable CAD, and no heart failure or known cancer diagnosis. Importantly, with longer follow-up and more events, the authors observed that NT-proBNP, previous atrial fibrillation, and absence of prior heart failure were independent predictors of a subsequent cancer diagnosis within 3 years. None of the evaluated factors predicted for cancer diagnosis after 3 years. This is an important study that adds identifies NT-proBNP as worthy of further study for validation and as a novel putative cancer biomarker in patients with CAD.

My specific comments are:

  1. The abstract methods should describe that this is a re-analysis of patients that were on a prospective trial

Thank you for the comment. We have now included this information in the abstract.

2. Why was 3 years chosen?

We have chosen 3 years because the number of patients diagnosed with cancer before and beyond 3 years was similar, thus providing a similar statistical consistency to the analyses performed in these two groups. If we had chosen an earlier point, the number of patients with cancer before 3 years would have been very low to yield consistent results. In the results section we have added a comment explaining this issue.

3. What was the median time/interval to developing cancer? (in the <3 yr and >3 yr cohorts)?

In the <3 years it was 1.53 (0.50-2.35) years while in the >3 year cohort it was 4.95 (3.12-9.28) years.

We have added these data to the new version of the paper. Thank you for this suggestion, as we believe that they may help to understand the results: It seems easy to understand that that a biomarker may anticipate cancer diagnosis in the next three years, but not five or nine years later.

4. While there wasn’t a statistically significant difference in the subgroup comparisons by histology for NT-proBNP levels (e.g. adeno vs carcinoma, lung/hem vs other), it would be really interesting to see the mean NT-proBNP levels for the various cancer sites/histologies in a table to assess trends. It might also be interesting to look at the squamous cell vs adeno subsets.

Given the low number of different types of cancer we are not sure that the looking at subgroups may provide consistent information. We must take into account that many cancers were represented by only one case. Those with the maximal prevalence were represented by 5-6 cases. We believe that our sample size is not large enough to make these analyses. They may be done in the future with large series. Of course we agree that NT-proBNP plasma levels may be more powerful to detect some types of cancers instead of all cancers, but we need to wait the publication of papers with large sample sizes.

5. Table 4 should include the HR/95% CIs/p-values for all the variables included in the model.

Thank you. We have added these data in Table 4.

Reviewer 2 Report

The paper investigates the predictors to predict cancer diagnosis in patients with coronary artery disease in almost 1000 patients.

Comments:

  • In the abstract, the authors want to predict cancer diagnosis in "short-term". However, a median follow-up of over 5 years (and 3 years in the multivariable model) is not short-term, this should be rephrased.
  • Furthermore, the authors state absence of prior heart failure as risk factor for cancer diagnosis. However, in the second sentence, the authors state they exluded patients with evidence of HF. Please clarify.
  • Hypothesis at the end of the introduction is missing. Please add. What is the puropse of the study?
  • Do the authors have follow-up blood samples? This would proof the robustness of the data. How did proBNP levels change in patients with cancer in this cohort? Please add and discuss.
  • The authors state: Multivariate Cox regression was performed including variables that showed a p value <0.20 at the univariate analysis. Why not p<0.05? Please explain.
  • Multi-variable analyses should be performed for different types of cancer. What type of cancer can be predicted using pro BNP?

Author Response

ANSWERS FOR REVIEWER 2

First of all we want to thank the reviewers and the editors for their comments that have undoubtedly contributed to improve the quality of the manuscript.

Below, we provide our answers to the reviewers.

1. In the abstract, the authors want to predict cancer diagnosis in "short-term". However, a median follow-up of over 5 years (and 3 years in the multivariable model) is not short-term, this should be rephrased.

Thank you. In the new version, we have chosen “middle-term” instead of “short-term”. We agree with you in that this describes better the findings of the study.

2. Furthermore, the authors state absence of prior heart failure as risk factor for cancer diagnosis. However, in the second sentence, the authors state they excluded patients with evidence of HF. Please clarify.

We agree that the terms used may be confounding. In the new version we explain that patients with heart failure at first visit were excluded. However we did not exclude patients with previous episodes of heart failure. In fact, about 12% of patients had previous heart failure.

In the new version, in the abstract we now say that the patients were free of cancer and decompensated HF at baseline. In addition, in the results section, at multivariate analysis, we emphasize that the variable predicting cancer diagnosis before three years is “Previous heart failure” instead of “Heart Failure”.

Thank you for this suggestion, as we believe that the paper is now easier to read.

3.Hypothesis at the end of the introduction is missing. Please add. What is the purpose of the study?

Although the hypothesis was explained in the first version, we have now rewritten it. Although we do not put the hypothesis at the end of the introduction, it has been placed in the line start of the text after a full-stop. We believe that it is now easier to the reader to get an idea of the objectives of this study.

4. Do the authors have follow-up blood samples? This would proof the robustness of the data. How did proBNP levels change in patients with cancer in this cohort? Please add and discuss.

No. We agree with you that this would have added value to the paper, but we do not have them. It would have been of great interest to re-assess proBNP levels at the moment the patients received a cancer diagnosis, for instance. We have added this point to the limitations section.

5. The authors state: Multivariate Cox regression was performed including variables that showed a p value <0.20 at the univariate analysis. Why not p<0.05? Please explain.

Both possibilities are right.

In studies when we get many variables with p<0.05 at the univariate analysis, we include only those variables, because if we use a cut-off of 0.20 we are going to include too many variables for the multivariate analysis.

On the other hand, in studies like this one, in which only a few variables (4  from a total of 37) analyzed are significant at univariate analysis, we prefer to use the cut-off of p<0.20 to avoid potential relevant information. For instance, at univariate analysis HF had a p=0.188, but at the multivariate analysis it turned out to be significant.

As we believe that this point is interesting, we include a short explanation in the Results section.

6. Multi-variable analyses should be performed for different types of cancer. What type of cancer can be predicted using pro BNP?

This is an interesting question, but we believe that it cannot be investigated in our study given the limited number of patients developing malignancies.

First, if we look at cancer localizations, we see that in many of them we only have 1 patient (Melanoma, pharynx, etc.). In fact the most frequent localizations are: prostate (5) and lung (5). With these numbers we cannot get solid conclusions

If we now look at the different histologic types of cancer (see section 3.3) we have 15 carcinomas and 11 adenocarcinomas and NT-proBNP were very similar (519.8+-706.5 pg/ml vs 480.4+-143.0 pg/ml, respectively; p=0.877). Then, again we believe that with these data, trying get more conclusions may lead to inconsistent information.

We think that we need to await the analysis of large series to investigate this aspect.

Round 2

Reviewer 2 Report

Dear authors,

thank you very much for the changes performed within the current revision of the mauscript.

However, it remains hard to conclude increased risk of cancer at 3 years following one blood sample at day 1. I think this limitation may not be addressed with this data.

Author Response

Dear authors,

thank you very much for the changes performed within the current revision of the manuscript.

It has been a pleasure. We believe that your comments have improved the manuscript.

However, it remains hard to conclude increased risk of cancer at 3 years following one blood sample at day 1. I think this limitation may not be addressed with this data.

We agree that having repeated determinations could have added consistency to the work. However, we have not plasma available during follow-up. We have stated this in the Limitations section.

Thank you very much